# A long term (1965-2015) ecological marine database from the LTER-Italy site Northern Adriatic Sea: plankton and oceanographic observations

Francesco Acri[1], Mauro Bastianini[1], Fabrizio Bernardi Aubry[1], Elisa Camatti[1], Alfredo Boldrin[1], Caterina Bergami[2], Daniele Cassin[1], Amelia De Lazzari[1], Stefania Finotto[1], Annalisa Minelli[3]*, Alessandro Oggioni[4], Marco Pansera[1], Alessandro Sarretta[5], Giorgio Socal[1], Alessandra Pugnetti[1]

[1] CNR-ISMAR, Arsenale - Tesa 104, Castello 2737/F, 30122 Venezia, Italy
[2] CNR-ISMAR, Via Gobetti 101, 40129 Bologna, Italy
[3] CNR-IRBIM, Largo Fiera della Pesca 2, 60125 Ancona, Italy
[4] CNR-IREA, Via Bassini 15, 20133 Milano, Italy
[5] CNR-IRPI, Corso Stati Uniti 4, 35127 Padova, Italy

*Correspondence to:* annalisa.minelli@gmail.com

## Abstract

In this paper we describe a 50 years (1965-2015) ecological database containing data collected in the Northern Adriatic Sea (NAS), one of the 25 research parent sites belonging to the Italian Long Term Ecological Research Network (LTER-Italy, http://www.lteritalia.it). LTER-Italy is a formal member of the international (https://www.ilter.network) and European (http://www.lter-europe.net/) LTER networks. The NAS is undergoing a process, led by different research institutions and projects, for the establishment of a marine ecological observatory, building on the existing facilities, infrastructures, and long-term ecological data. Along this process, the implementation of the Open Access and Open Science principles has started, by creating an open research lifecycle that involves sharing ideas and results (scientific papers), data (raw and processed), metadata, methods, and software. The present data paper is framed within this wider context. The database is composed of observations on abiotic parameters, phyto- and zooplankton abundances, collected during 299 cruises in different sampling stations, in particular in the Gulf of Venice: we describe here the sampling and analytical activities, the parameters, and the structure of the database. The database is available at http://doi.org/10.5281/zenodo.3516717 (Acri et al., 2019), it was also uploaded in the DEIMS-SDR repository (Dynamic Ecological Information Management System - Site and Dataset Registry, https://deims.org/), which is the official sites and data registry for LTER International network.

## 1. Introduction

We describe in this paper a 50 years (1965-2015) ecological database containing data on plankton communities and related abiotic parameters, collected in the Northern Adriatic Sea (NAS). Plankton communities, which are at the base of aquatic ecosystem functioning, have a broad and diversified range of seasonal patterns, multi-annual trends, and shifts across different marine ecosystems: making available long term series of plankton and oceanographic observations provides unique and precious tools for depicting reliable patterns of average annual cycles and for detecting significant changes and trends in response to global or local pressures and impacts.

Open Data is nowadays considered a crucial issue in both scientific research and public administration and management. Wilkinson et al. (2016) conceived the "FAIR" data management principles, which states that data must be "Findable, Accessible, Interoperable and Reusable". The open access to data is one crucial step of Open Science (http://www.budapestopenaccessinitiative.org/read, European Commission, 2016), which is a wider approach embracing transparency at all stages of the research process, from research ideas to papers, open access to data, codes, and software. Open Science is actually a democratic way of making freely available, for every researcher and stakeholder, research ideas, data, metadata, tools, and outcomes. From the researcher point of view, open practices have been reported to give advantage, first of all, to open new frontiers in science (Science|Business network's cloud consultation group, 2019) and provide solutions to urgent societal problems (Palen et al., 2015; Tai and Robinson, 2018); moreover, it allows gaining more citations, media

attention, potential collaborators, and funding opportunities (Eisenbach, 2006; McKiernan et al., 2016, Tennant et al., 2019) and it is vital for leaving a heritage to future generations.

Ecology, being a multidisciplinary science, can surely benefit from the Open Science approach, which is, however, a matter of interest and discussion among ecologists only since the last decade  (Reichman et al., 2011). Yet, the cultural shift from "data ownership to data stewardship" is not widely accomplished and data sharing standards, both from a technical and ethical point of view, have just started to be established (Hampton et al., 2015).

The Open Science approach is fostered in the data management plans of the Long Term Ecological Research (LTER) networks, at the national, European (LTER-Europe: http://www.lter-europe.net/) and global level (International LTER, ILTER: https://www.ilter.network),  being considered a crucial step to advance socio-ecological research and education (Mirtl et al., 2018). ILTER provides a globally distributed network of long-term research sites for multiple purposes and uses in the fields of ecosystem, biodiversity, and socio-ecological research,  it currently consists of 44 national networks, managing more than 700 sites worldwide (Haase et al., 2018; Mirtl et al., 2018). LTER-Italy (www.lteritalia.it), a formal component of ILTER and LTER Europe since 2006, consists of 79 research sites, organized in 25 parent sites, which include terrestrial, freshwater, transitional and marine ecosystems, managed and coordinated by public research, monitoring Institutions and Universities (Bergami et al., 2019).

The LTER marine component, which represents around 10% of global ILTER sites, focuses mainly on ecosystem structure and function, in response to a wide range of environmental forcing factors, using long-term, site-based research. As a result of the wide range and of the exceptional rate and intensity of human impacts, the scientific value of long-term ecological observations is more crucial than ever for effective assessment, management, and prediction of the state and pressure in the marine environment. The creation and maintenance of marine ecological observatories, able to arrange and maintain integrated, harmonized and coherent long-term ecological observations, is actually stressed as a relevant step at the European level, for sustaining European marine policies (Benedetti-Cecchi et al., 2018; European marine Board 2019).

The marine component of LTER-Italy is made up of eight parent sites, mainly representing transitional and coastal ecosystems. Among them, the NAS is a significant geographical zone for the establishment of a marine ecological observatory, due to the concomitant presence of sensitive habitats, numerous ongoing monitoring, and research activities, as well as of heavy and diversified human pressures and economic interests. For these main reasons, during the years 2017-18, the Italian national flagship project RITMARE ("Italian research for the sea", http://www.ritmare.it/), funded by the Italian Ministry of University and Research, dedicated a Research Line to the establishment of a marine ecological observatory in the NAS. Building on the existing facilities, infrastructures and long-term ecological data, it aims at enhancing the marine observational capacities and at activating synergies among the main conservation management questions and key ecological and oceanographic variables. Along this process, it appeared crucial to start applying the Open Science principles, by creating an open research lifecycle, which foresees sharing each step of the process, from ideas and results (scientific papers) to data (raw and processed), from metadata to methods and software. The ideas and plans for the development of the Open Science principles to the NAS ecological observatory, which we named project "EcoNAOS" (Ecological Northern Adriatic Open Science Observatory System), are thoroughly described by Minelli et al. (2018).

This data paper represents one relevant step of this wider activity. The database that we present is composed of observations on abiotic (physical and chemical) parameters and phyto- and zooplankton abundances, collected in 50 years (from 1965 to 2015), during cruises which interested different sampling stations across the NAS, in particular in the Gulf of Venice. Here we describe the sampling and analytical activities, the parameters, and the structure of the database.

2. **The LTER-Italy parent site Northern Adriatic Sea**

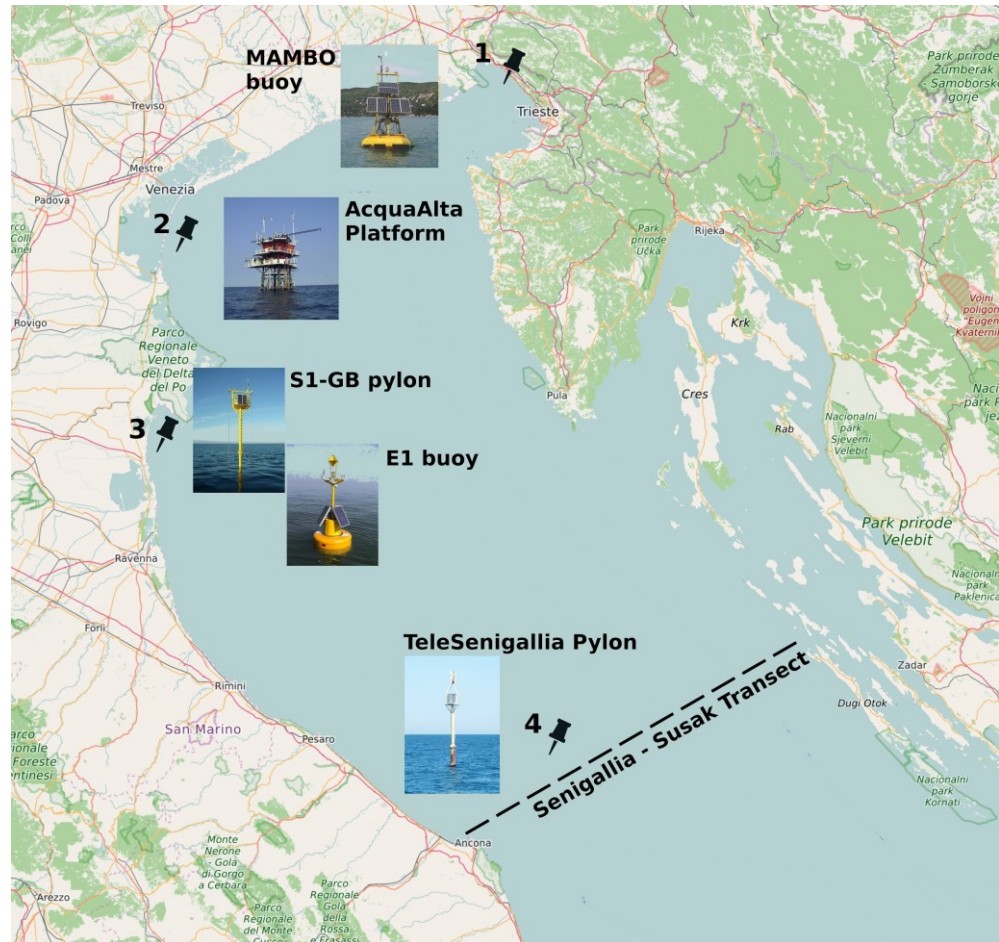

Figure 1 - The LTER-Italy parent site Northern Adriatic Sea, with its four research sites. 1: Gulf of Trieste; 2: Gulf of Venice; 3: Po Delta and Romagna Coast; 4: Senigallia-Susak Transect. The fixed point observatories at each research site are evidenced (see Ravaioli et al., 2016 for a full description). Base map credits: © OpenStreetMap contributors 2019. Distributed under a Creative Commons BY-SA License.

The NAS (Figure 1) is the northernmost basin of the Mediterranean Sea and one of its most productive areas. It is characterized by a shallow depth and by a dominant cyclonic circulation. The oceanographic and meteorological parameters show a marked seasonal and interannual variability. The major forcings of the system are represented by the remarkable river inputs along the Italian coast, the Eastern Adriatic Current-EAC, which brings high salinity and oligotrophic waters from the southern basin, and the notable sea-level range, relatively to the rest of Mediterranean area. The urban and industrial inputs and the hydrodynamic exchange between the NAS and the lagoons located along the Italian coast are also elements of ecological relevance. A trophic gradient, decreasing from northwest to southeast, is typically observed in the basin, in which the nutrient-rich waters coming from the rivers are mainly spread southward and eastward from the Italian coast (Bernardi Aubry et al., 2006; Solidoro et al., 2009). The NAS is subject to multiple anthropogenic impacts (e.g., nutrient inputs, coastal urbanization, fishing activity, tourism, and maritime trade). The basin has undergone overfishing (Fortibuoni et al., 2010), marked eutrophication (during the 70s; Giani et al., 2012), followed by a phase of oligotrophication (years 2000s; Mozetič et al., 2010) and by a recent increase of nutrient concentrations (since 2007; Totti et al., 2019). The NAS has also been subjected to frequent development of mucilage aggregates (Giani et al., 2005; De Lazzari et al., 2008), until the first decade of the 2000s.

The LTER-Italy parent site NAS includes four research sites (Gulf of Trieste, Gulf of Venice,  Po Delta and Romagna Coast, Senigallia-Susak Transect; Figure 1), where meteo-oceanographic and biological data, mainly on plankton (Table 2), are gathered both during oceanographic cruises and at fixed point observatories. Detailed information can be found in the ILTER Dynamic Ecological Information Management System Site and Dataset Registry, DEIMS-SDR (https://deims.org/92fd6fad-99cd-4972-93bd-c491f0be1301) (Wohner et al., 2019). The database we describe here refers to an area of about 40000 km$^2$, ranging between  43.7° and 45.8° North and 12.2° and 14.3° East (coordinate reference system: WGS84).

**3. Description of the database**

The database described in this data paper is composed of 108687 records. Each record is intended as a timestamped and
georeferenced set of information, individuated by a row in the database. These observations belong to 22 datasets coming from
299 oceanographic cruises, carried out from 1965 to 2015.

Due to the long time coverage, the collection and analysis system for many parameters changed in time, thus making the
database very heterogeneous for what concerns data management and organization. The heterogeneity is mainly due to:

● Sampling frequency: e.g., data coming from CTD (Conductivity, Temperature, Depth) sensors, such as temperature,
oxygen, and pH, are registered in real-time at each meter in depth; other parameters, like nutrients and phytoplankton,
are sampled at a lower time-frequency and at variable depths. The overall depth coverage ranged between 0-63 m,
the sampling frequency from monthly to seasonal;
● Data treatment: some data are basically raw, e.g., data registered by CTD are reported into the database as they are
delivered from the instrument; other data need some elaboration to obtain specific parameters' value (e.g., nutrients,
chlorophyll-*a*, plankton abundance);
● Methodologies and units of measurements: e.g., changes of methodologies due to the introduction of CTD
measurements; change of the units of measure of salinity, which passed from g l$^{-1}$ to a dimensionless parameter.
● Data format: data collected between 1965 and 1990 were registered only on paper archives, while those from 1990
onwards on spreadsheets.

In particular, methodological protocols and associated documentation changed through time. Several sensors are described and
extensively documented through the GET-IT platform (Geoinformation Enabling ToolkIT starterkit®, see Section 5), where
it is possible to visualize all the observations related to a specific instrument or method. Other protocols have undergone a
deep metadatation process by analyzing ancillary historical metadata (Scovacricchi, 2017). In this case, it is not immediately
possible to obtain data related to a specific protocol, but it is still possible to filter data by method by importing the .csv file in
a spreadsheet.

**3.1 Data sources and geographical coverage**
Data sources for this database come mainly from oceanographic cruises that were carried out on 12 different research vessels,
at the basin scale (Table 1). The other observations come from sampling stations located next to the fixed automatic sensors:
in this case the cruises are named as the nearby sensor, i.e.: 576 observations at the Paloma buoy (Gulf of Trieste), 1284 at the
Acqua Alta oceanographic tower (Gulf of Venice),  138 at the S1 buoy (Po Delta). The data were gathered in the frame of
many different projects that are all mentioned in the database:

| Operation period | Research Vessel (R/V) | Nr. of observations |
|---|---|---|
| 1965-1966 | Vercelli | 861 |
| 1966 | Sea Quinn | 60 |
| 1966-1990 | Bannock | 997 |
| 1968-2002 | D'Ancona | 45357 |
| 1977 | Marsili | 23 |
| 1979-1980 | Mysis | 48 |
| 1979-1990 | Vila Vilebita | 139 |
| 1986-1988 | Minerva | 737 |
| 2003 | Boreana | 2158 |
| 2003-2015 | Dallaporta | 43689 |
| 2007-2015 | Litus | 1900 |
| 2012-2014 | Urania | 12718 |

Table 1 – Operation periods of the different research vessels between 1965 and 2015 and number of observations.

Until the early 1990s, GPS systems were not usually on board of research vessels. For this reason, oceanographers used to
refer to a fixed grid covering the entire research area and identified the sampling positions (stations) with the nodes of this
grid. An example of grids used for this purpose is reported in Figure 2 (Franco, 1972).
In Figure 3a, the geographical coverage of the entire database is shown. Red dots represent the real observation points, while
the nodes of the grid are evidenced with black crosses. Observations referring to a specific station were assigned to the
coordinates of the corresponding node on the grid even if the real position was not precisely located on the grid node. This
resulted in a cloud of points in the nearby of each sampling station. Since our main aim was to preserve most of the information
for each observation, we decided not to "correct" the position of these points (see an example in Figure 3b for the station
09/0E).
In the following years, when the GPS allowed a better precision of the sampling position, researchers often continued referring
to the nodes of the grid for the station names and they adopted a nomenclature coherent with the one of the original grid also
for new sampling stations. For example, the new sampling point located eastward of the "09/2E" station is named "10/2E",
since it is located at the same longitude (2E), but different latitude of "09/2E" station (Figure 3b). In Figure 3c, a 3D view of
the entire database is shown.
Due to transcription errors occurred during the oldest cruises, some data were misplaced, falling on land or outside the NAS.
A Python script (available under GNU GPL v.3 license here: https://github.com/CNR-ISMAR/econaos/tree/master) has been
written in order to correct this kind of errors. The same script implemented also a routine to homogenize different names of
the same sampling station (e.g. station "020D" could appear as well as station "02-0D" or "02/0D" or "020D_07/07/1968").
We selected the name reported on the original stations' network grid (Figure 2) and we created from these stations a vector
layer (black crosses in Figure 3). Finally, since some stations changed their name through time, in order to maintain coherence
with the same sampling point, we appointed them with the last, most recently used name.

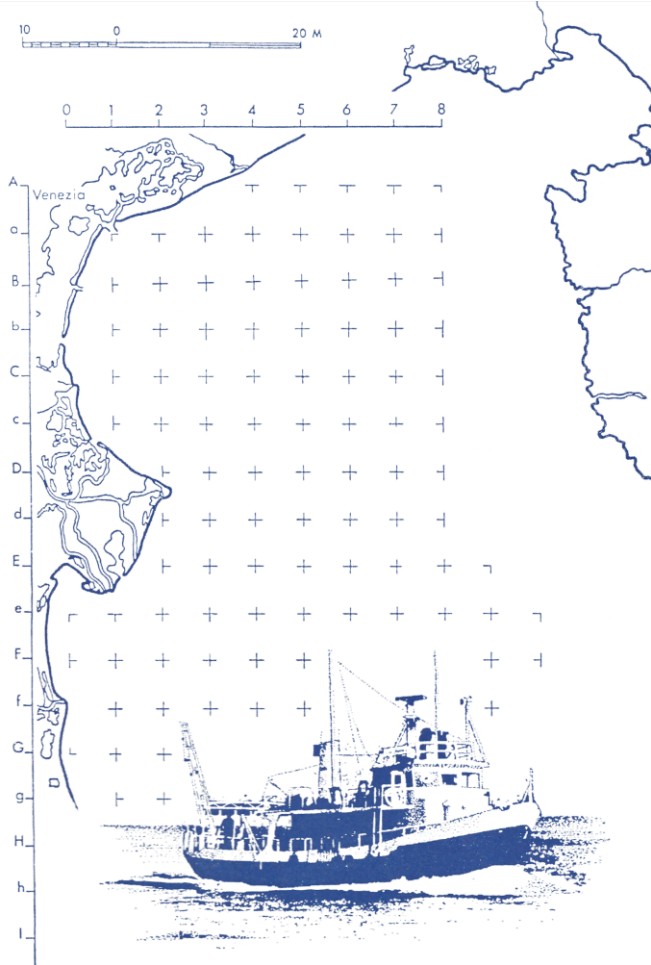

Figure 2 - An example of sampling stations based on the regular grid created in the NAS for the cruises from 1966 to 1980
(from Franco, 1972).

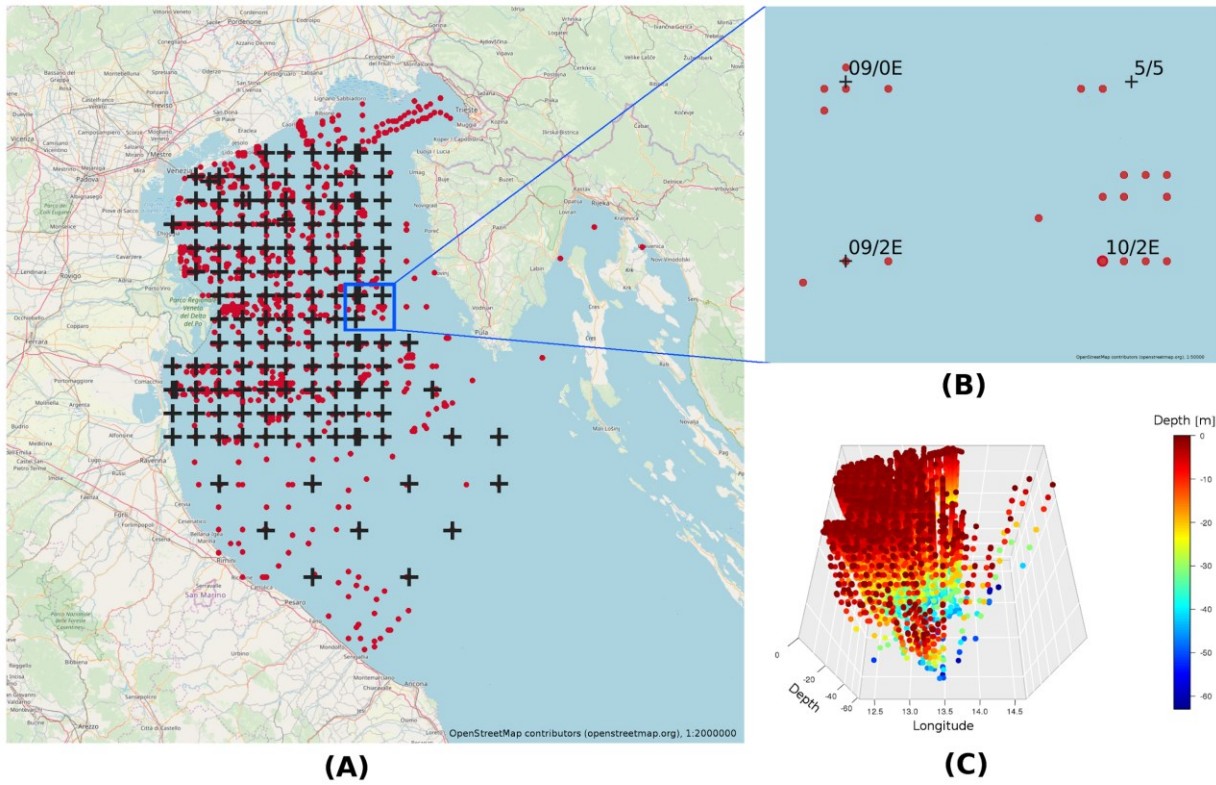

Figure 3 - (A) Geographical distribution of the observations: red dots for observations; black crosses for nodes of the grid. (B)
Example of cloud distribution of observations around sampling station 09/2E and the naming of new sampling station 10/2E.
(C) 3D view of the database. Base map credits: © OpenStreetMap contributors 2019. Distributed under a Creative Commons
BY-SA License.

**3.2 Parameters: history, time coverage, and sensors**

Samples collected during each cruise, whatever the station of collection, were then analyzed in the laboratory by means of
diverse techniques. Since 2000 analytical quality of nutrients and chlorophyll analyses is assessed through participation to the
Quality Assurance of Information for Marine Environmental Monitoring In Europe (QUASIMEME;
http://www.quasimeme.org) international laboratory proficiency-testing. The complete list of the parameters of the database
is reported in Table 2, together with some descriptive elements, i.e.:

● Total number of observations,
● Temporal coverage (from the first to the last record),
● Method or sensor currently used,
● Current unit of measure.

| Parameter | Number of observations | Temporal coverage | Current sensor | Unit of measure | Acronym in the database |
|---|---|---|---|---|---|
| Transparency | 2322 | 1965-2015 | Secchi Disk | m | Secchi |
| Temperature | 107648 | 1965-2015 | CTD | C | Temp |
| Salinity | 107655 | 1965-2015 | CTD | dimensionless | Sal |
| Density anomaly | 99961 | 1965-2015 | Derived from temperature and salinity | kg m$^{-3}$ | Dens |
| pH | 70376 | 1965-2011 | pH glass membrane and pH electrode | pH unit | pH |

| | | | | | |
|---|---|---|---|---|---|
| Alkalinity | 492 | 1965-2002 | Titrino titration | meq $l^{-1}$ | Alky |
| Oxygen | 12791 | 1965-2012 | Oxygen Polarographic sensor | cc $l^{-1}$ | Oxyg (ml/l) |
| N-NH3 | 11154 | 1965-2015 | Automated nutrient analysis | $\mu$m $dm^{-3}$ | NH3 (microMol) |
| N-NO2 | 11232 | 1965-2015 | Automated nutrient analysis | $\mu$m $dm^{-3}$ | NO2 (microMol) |
| N-NO3 | 11299 | 1965-2015 | Automated nutrient analysis | $\mu$m $dm^{-3}$ | NO3 (microMol) |
| P-PO4 | 11191 | 1965-2015 | Automated nutrient analysis | $\mu$m $dm^{-3}$ | PO4 (microMol) |
| Si-SiO4 | 11420 | 1965-2015 | Automated nutrient analysis | $\mu$m $dm^{-3}$ | Si (microMol) |
| Chlorophyll-*a* | 11541 | 1965-2015 | Spectrofluorimeter | $\mu$g $l^{-1}$ | Chla (ug/l) |
| Pheopygments | 6352 | 1979-2015 | Spectrofluorimeter | $\mu$g $l^{-1}$ | Pheo (ug/l) |
| Total Phytoplankton | 3463 | 1977-2015 | Inverted microscope | Cells $l^{-1}$ | Phyto TOT (cell/ml) |
| Diatoms | 3070 | 1977-2015 | Inverted microscope | Cells $l^{-1}$ | Diato (cell/ml) |
| Dinoflagellates | 3070 | 1977-2015 | Inverted microscope | Cells $l^{-1}$ | Dino (cell/ml) |
| Coccolithophores | 3070 | 1977-2015 | Inverted microscope | Cells $l^{-1}$ | Cocco (cell/ml) |
| Others | 3070 | 1977-2015 | Inverted microscope | Cells $l^{-1}$ | Flag (cell/ml) |
| Total Zooplankton | 372 | 1987-2015 | Stereo microscope | Ind. $m^{-3}$ | Zoo (ind/m^3) |

Table 2 - Database parameters and main descriptive information.

Instruments and sensors changed over the 50 year period, due to technological and scientific progress. Furthermore, instruments are also subject to degradation and need to be replaced. It is essential to preserve the information about these instrument changes and upgrading, to track the reliability of the measurements.

In order to appropriately document data and guarantee the consistency of data within the database, we collected most ancillary information as possible on the changes occurred in time for each parameter measurement. To this purpose, a thorough review of historical sources (e.g. logbooks and manual transcription in spreadsheets) was carried out (Scovacricchi, 2017), working in cooperation with some researchers - now retired - who participated to the first cruises and referring as well to papers by Franco (1970, 1972 and 1982), which describe methods and instruments during a number of oceanographic cruises in the NAS from 1965 to 1979.

Plankton data are particularly sensitive to the skill of the operators, in particular during the microscope analyses of the samples. The change of the operators, which necessarily occurred during 50 years, actually could hamper the data comparison across time. To deal with this issue, internal education and recurring calibration of taxonomic competence were carefully considered, with training periods and intercalibrations phases.

Since 2006 the taxonomic revision of the phytoplankton species has been made according to "Algaebase" (www.algaebase.org), the global algal database of taxonomic, nomenclatural and distributional information for the zooplankton the Marine Planktonic Copepods catalog (https://copepodes.obs-banyuls.fr/en/links.php, Razolus et al., 2005-2019) has been used. In the past, for phyto- and zooplankton analyses several texts and monographs were used (Berard-

220 Therriault et al., 1999; Harris et al., 2000; Heimdal, 1993; Hendey, 1964; Hustedt, 1930-1966; Pascher, 1915; Peragallo and
221 Peragallo, 1897-1908; Rampi and Bernhardt, 1980; Schiller, 1931-37; Throndsen, 1993; Tomas, 1997).
The phytoplankton was gathered and analyzed with the same method (Utermohl, 1958) across the years. In the database we
report the total phytoplankton abundances and the following main groups: diatoms, dinoflagellates (naked and armoured cells),
coccolithophorids and "others", which include the sum of cells belonging to cryptophyceans, crysophyceans,
prymnesiophyceans (except coccolithophorids), prasinophyceans and chlorophyceans, whose sizes lie between 4 and 20 μm
and often remain undetermined. Mesozooplankton was always identified under a stereo-microscope and expressed as the total
number of organisms per cubic meter. Compared to phytoplankton, the mesozooplankton data are much fragmented over time:
they cover a 28 year period, from 1987 to 2015, for a total of 372 observations.

**4. Database structure and analysis**

The present version of the database is recorded in a unique spreadsheet (Figure 4), carrying information, for each record, about:
● Coordinates (longitude-latitude) of the sampling station;
● Sampling depth;
● Sampling station name;
● Cruise and R/V (Ship) name;
● Sampling date and time;
● Water column depth (Bot. Depth);
● Instrument/method used for each measurement and relative parameter value.

| Long | Lat | Depth | Station | Cruise | Ship | YYYY-MM-DD | hh:mm:ss | Bot. Depth | Temp_sensor | Temp | Sal_sensor | Sal |
|---|---|---|---|---|---|---|---|---|---|---|---|---|
| 12.68 | 45.33 | 0.5 | B | PP/1 | VERCELLI | 1965-04-12 | 9:33:00 | 23 | Tilting thermometer | 13.12 | Morh Knudsen titration | 29.61 |
| 12.68 | 45.33 | 5 | B | PP/1 | VERCELLI | 1965-04-12 | 9:33:00 | 23 | Tilting thermometer | 12.35 | Morh Knudsen titration | 35.66 |
| 12.68 | 45.33 | 10 | B | PP/1 | VERCELLI | 1965-04-12 | 9:33:00 | 23 | Tilting thermometer | 12.45 | Morh Knudsen titration | 35.43 |
| 12.68 | 45.33 | 20 | B | PP/1 | VERCELLI | 1965-04-12 | 9:33:00 | 23 | Tilting thermometer | 12.14 | Morh Knudsen titration | 38.01 |
| 12.86 | 45.28 | 0.5 | C | PP/1 | VERCELLI | 1965-04-12 | 12:20:00 | 29 | Tilting thermometer | 12.25 | Morh Knudsen titration | 35.44 |
| 12.86 | 45.28 | 5 | C | PP/1 | VERCELLI | 1965-04-12 | 12:20:00 | 29 | Tilting thermometer | 12.24 | Morh Knudsen titration | 35.46 |
| 12.86 | 45.28 | 10 | C | PP/1 | VERCELLI | 1965-04-12 | 12:20:00 | 29 | Tilting thermometer | 11.16 | Morh Knudsen titration | 37.79 |
| 12.86 | 45.28 | 20 | C | PP/1 | VERCELLI | 1965-04-12 | 12:20:00 | 29 | Tilting thermometer | 12.3 | Morh Knudsen titration | 37.92 |
| 12.48 | 45.40 | 0.5 | A | PP/2 | VERCELLI | 1965-04-28 | 6:42:00 | 16.4 | Tilting thermometer | 12.27 | Morh Knudsen titration | 33.04 |
| 12.48 | 45.40 | 1 | A | PP/2 | VERCELLI | 1965-04-28 | 6:42:00 | 16.4 | Tilting thermometer | 12.37 | Morh Knudsen titration | 33.39 |
| 12.48 | 45.40 | 5 | A | PP/2 | VERCELLI | 1965-04-28 | 6:42:00 | 16.4 | Tilting thermometer | 12.44 | Morh Knudsen titration | 35.39 |
| 12.48 | 45.40 | 10 | A | PP/2 | VERCELLI | 1965-04-28 | 6:42:00 | 16.4 | Tilting thermometer | 12.23 | Morh Knudsen titration | 37.3 |
| 12.68 | 45.33 | 0.5 | B | PP/2 | VERCELLI | 1965-04-28 | 9:10:00 | 22.3 | Tilting thermometer | 12.49 | Morh Knudsen titration | 32.9 |
| 12.68 | 45.33 | 5 | B | PP/2 | VERCELLI | 1965-04-28 | 9:10:00 | 22.3 | Tilting thermometer | 12.43 | Morh Knudsen titration | 33.78 |
| 12.68 | 45.33 | 10 | B | PP/2 | VERCELLI | 1965-04-28 | 9:10:00 | 22.3 | Tilting thermometer | 11.92 | Morh Knudsen titration | 37.21 |
| 12.68 | 45.33 | 20 | B | PP/2 | VERCELLI | 1965-04-28 | 9:10:00 | 22.3 | Tilting thermometer | 10.5 | Morh Knudsen titration | 37.72 |
| 12.86 | 45.28 | 0.5 | C | PP/2 | VERCELLI | 1965-04-28 | 11:20:00 | 31 | Tilting thermometer | 12.4 | Morh Knudsen titration | 34.2 |
| 12.86 | 45.28 | 5 | C | PP/2 | VERCELLI | 1965-04-28 | 11:20:00 | 31 | Tilting thermometer | 12.09 | Morh Knudsen titration | 36.15 |
| 12.86 | 45.28 | 8 | C | PP/2 | VERCELLI | 1965-04-28 | 11:20:00 | 31 | Tilting thermometer | 11.5 | Morh Knudsen titration | 37.38 |
| 12.86 | 45.28 | 20 | C | PP/2 | VERCELLI | 1965-04-28 | 11:20:00 | 31 | Tilting thermometer | 10.42 | Morh Knudsen titration | 37.9 |
| 12.48 | 45.40 | 0.5 | A | PP/3 | VERCELLI | 1965-05-13 | 6:47:00 | 16 | Tilting thermometer | 15.92 | Morh Knudsen titration | 33.66 |
| 12.48 | 45.40 | 1 | A | PP/3 | VERCELLI | 1965-05-13 | 6:47:00 | 16 | Tilting thermometer | 15.8 | Morh Knudsen titration | 33.77 |
| 12.48 | 45.40 | 5 | A | PP/3 | VERCELLI | 1965-05-13 | 6:47:00 | 16 | Tilting thermometer | 14.92 | Morh Knudsen titration | 33.51 |
| 12.48 | 45.40 | 10 | A | PP/3 | VERCELLI | 1965-05-13 | 6:47:00 | 16 | Tilting thermometer | 11.34 | Morh Knudsen titration | 37.61 |
| 12.68 | 45.33 | 0.5 | B | PP/3 | VERCELLI | 1965-05-13 | 9:23:00 | 21 | Tilting thermometer | 17.4 | Morh Knudsen titration | 33.84 |
| 12.68 | 45.33 | 5 | B | PP/3 | VERCELLI | 1965-05-13 | 9:23:00 | 21 | Tilting thermometer | 15.66 | Morh Knudsen titration | 36.2 |
| 12.68 | 45.33 | 10 | B | PP/3 | VERCELLI | 1965-05-13 | 9:23:00 | 21 | Tilting thermometer | 13.64 | Morh Knudsen titration | 37.3 |
| 12.68 | 45.33 | 20 | B | PP/3 | VERCELLI | 1965-05-13 | 9:23:00 | 21 | Tilting thermometer | 11.83 | Morh Knudsen titration | 37.72 |
| 12.86 | 45.28 | 0.5 | C | PP/3 | VERCELLI | 1965-05-13 | 12:15:00 | 31 | Tilting thermometer | 18.03 | Morh Knudsen titration | 33.01 |

Figure 4 - An example of the database showing the fields for each observation.

Around 89% of the observations of the database refers to the years 1999-2015 and the remaining 11% covers the previous 33
244 years (see Figure 5a for details). This is mainly due to the adoption of CTD probes since 1999 for measuring abiotic parameters
at each meter depth, leading to an imbalance between the observations before (e.g. 778 in 1978) and after 1999 (e.g. 11359 in
2004)In Figure 5b observations from oceanographic cruises onboard of the different research vessels are shown (see also Table
1). The number of observations on abiotic parameters (nutrients, alkalinity, and transparency) is higher than the biotic
(chlorophyll-*a*, phytoplankton, and zooplankton abundances) ones up to the year 2000; since then, they become comparable
(Figure 5c).
The database presents a heterogeneous number of observations for each parameter, mainly due to: (i) parameter priority for
the specific research conducted, (ii) the instruments and analytical efforts required, and (iii) the specific funding programs and
resources.

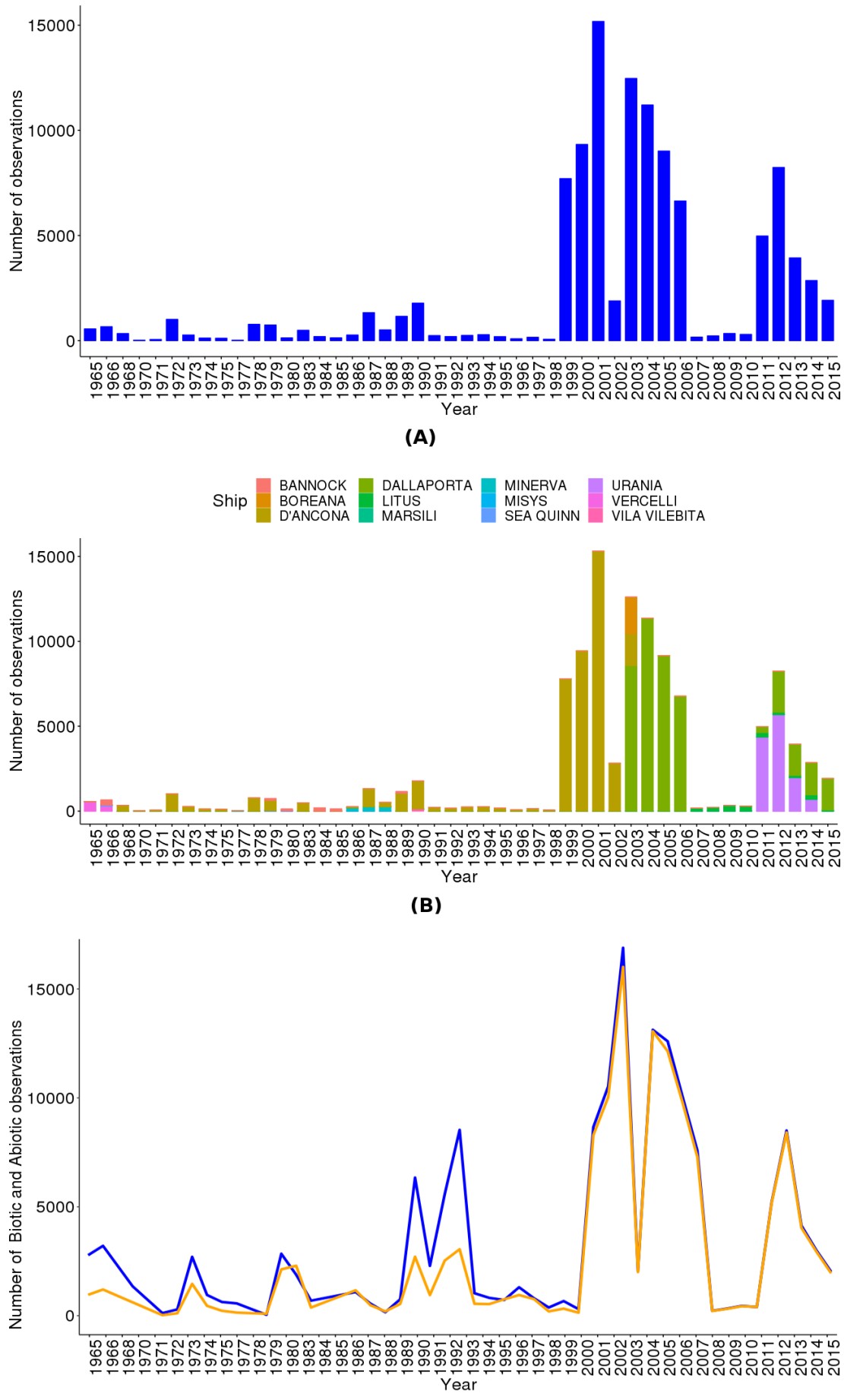

Figure 5 - Total number of observations over the whole period (A), clustered by research vessel (B), and by biotic (orange
line) and abiotic (blue line) parameters (C).

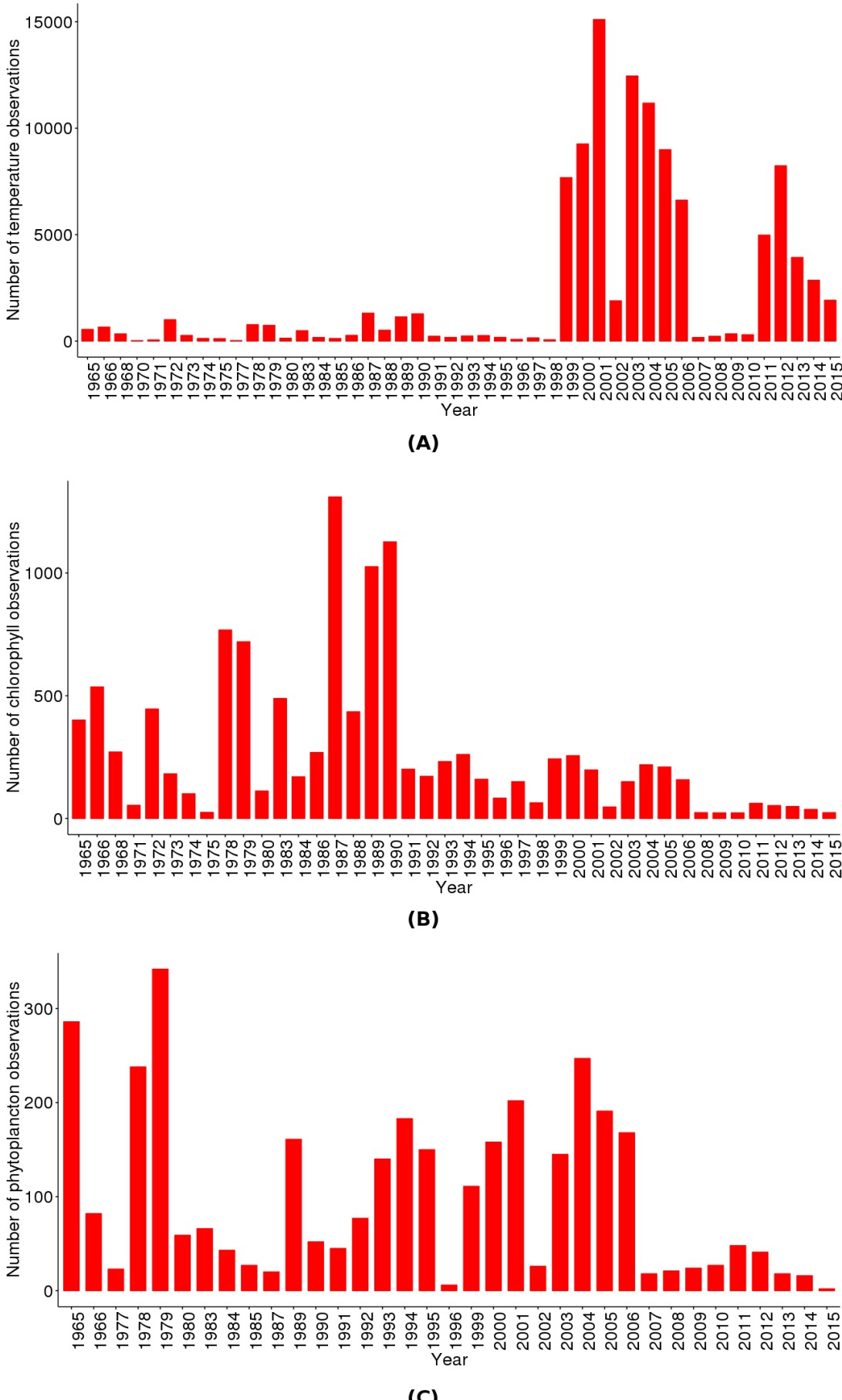

**(A)**

**(B)**

**(C)**

Figure 6 - Distribution over the years of the temperature (A), chlorophyll-*a* (B) and phytoplankton (C) observations

In Figure 6 we compare the total number of observations of one physical (temperature) and two biological (chlorophyll-*a* and phytoplankton abundance) parameters. All the three parameters were measured each year, although with different frequency. Temperature attains up to ~15000 records, while chlorophyll-*a* ~1200 records at most and phytoplankton ~300. The number of temperature data has a temporal distribution similar to the general one described in Figure 5a, where 89% of the observations occurred in the last 17 years, due to the adoption of CTD probes. Chlorophyll-*a* observations show instead peaks during the years 1987-1990, due to intense regional monitoring activities occurring in those years. The lowest number of phytoplankton observations is mainly due to the complex and time-consuming analytical procedure, which do not allow processing too many samples, and to the reduction of extensive monitoring activities since 2006.

## 5. Data visualization

The data management activities of the national flagship project RITMARE (Fugazza et al. 2014) allowed to develop two tools to enhance the deployment of a distributed Spatial Data Infrastructure (SDI) for Italian marine researchers community. SDI is an interoperable technological infrastructure for preservation, publication,and discovery of geospatial, modeled on standard (Open Geospatial Consortium - OGC, World Wide Web Consortium - W3C, and INSPIRE Directive 2007/2/EC) web services. In order to strengthen the RITMARE infrastructure, the Open Source software suite GET-IT (Oggioni et al., 2017; Menegon et al., 2017) and the customizable, template-driven metadata editor EDI (Pavesi et al. 2016; Tagliolato et al. 2016; https://github.com/SP7-Ritmare/EDI-NG_client) have been developed and released as Open Source code. One of the nodes of the distributed SDI provides geospatial data collected by CNR-ISMAR marine researchers (http://vesk.ve.ismar.cnr.it). Following the OGC Sensor Web Enablement (SWE) web service, each instrument or procedure has to be filled out as a "sensor", then observations can be provided, for a specific parameter, as OGC O&M (Observations and Measurements) web standards. Through the EDI interface, integrated within GET-IT software suite, a first core of sensors was already tested and uploaded in 2015 (Bastianini et al. 2015). A number of buoys (e.g. ABATE - Seabird SBE 19 Plus V2), laboratory instruments (e.g. Spettrophotometer Perkin Elmer), methods (e.g. Titration Winkler) and sensors, have been described for this study by using XML SensorML v2.0 language and their metadata, including manufacturer (provided as RDF, which stays for Resource Description Framework, Friends Of A Friends FOAF in Oggioni, 2019), owner and operator contacts, measured parameters, position, documentation, and history, can be easily visualized in separate dedicated landing pages (Figure 7). Currently, in the CNR-ISMAR GET-IT data node, 35 sensors have been described (http://vesk.ve.ismar.cnr.it/sensors/), for which it is possible to upload observations, collected from different stations in the NAS.

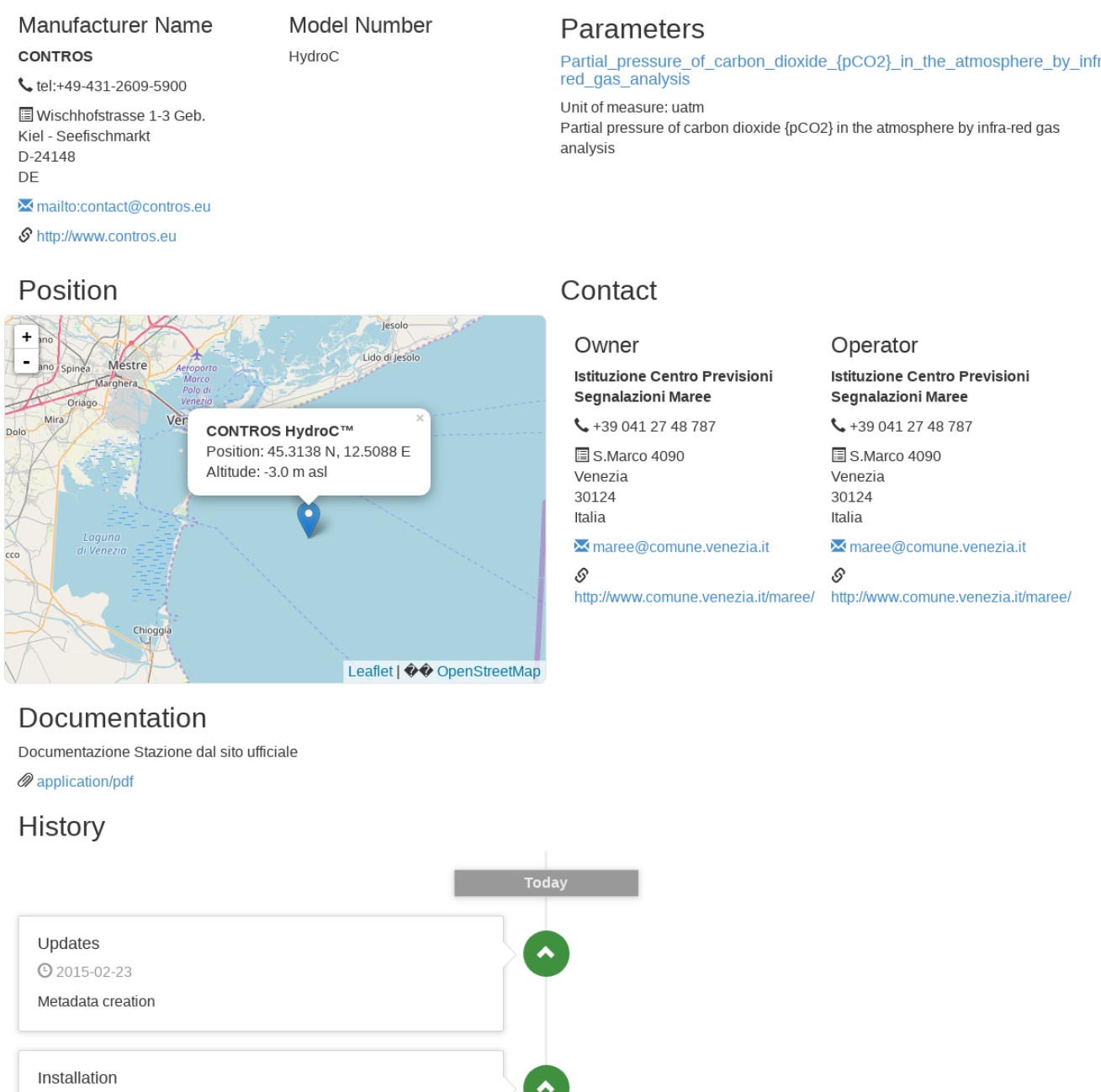

# CONTROS HydroC™ (Piattaforma Acqua Alta - CONTROS HydroC)

The CONTROS HydroC™ CO2 sensor is a unique underwater dioxide sensor for in-situ and online measurements of dissolved CO2. The versatile HydroC™ CO2 is suitable for platform installations (e.g. ROV s or AUV s), long-term deployments (e.g. buoys and moorings) as well as for profiling applications (e.g. water sampling rosettes). Fields of application include: ocean acidification research, climate studies, air-sea gas exchange, limnology, fresh water control, aquaculture/fish farming, carbon capture and storage – monitoring, measurement and verification (CCS-MMV).

## Manufacturer Name

**CONTROS**

📞 tel:+49-431-2609-5900

🗒 Wischhofstrasse 1-3 Geb.
Kiel - Seefischmarkt
D-24148
DE

✉ mailto:contact@contros.eu

🔗 http://www.contros.eu

## Model Number

HydroC

## Parameters

Partial_pressure_of_carbon_dioxide_{pCO2}_in_the_atmosphere_by_infra-red_gas_analysis

Unit of measure: uatm
Partial pressure of carbon dioxide {pCO2} in the atmosphere by infra-red gas analysis

## Position

CONTROS HydroC™
Position: 45.3138 N, 12.5088 E
Altitude: -3.0 m asl

Leaflet | ◆◆ OpenStreetMap

## Contact

### Owner

**Istituzione Centro Previsioni Segnalazioni Maree**

📞 +39 041 27 48 787

🗒 S.Marco 4090
Venezia
30124
Italia

✉ maree@comune.venezia.it

🔗 http://www.comune.venezia.it/maree/

### Operator

**Istituzione Centro Previsioni Segnalazioni Maree**

📞 +39 041 27 48 787

🗒 S.Marco 4090
Venezia
30124
Italia

✉ maree@comune.venezia.it

🔗 http://www.comune.venezia.it/maree/

## Documentation

Documentazione Stazione dal sito ufficiale

📎 application/pdf

## History

Today

Updates
🕐 2015-02-23
Metadata creation

Installation
🕐 2014-05-01
Date of installation

Figure 7 - Example of the sensor description provided by GET-IT. Information about manufacturer, owner and operator contacts, measured parameters, position, documentation, and history are displayed. Base map credits: © OpenStreetMap contributors 2019. Distributed under a Creative Commons BY-SA License.

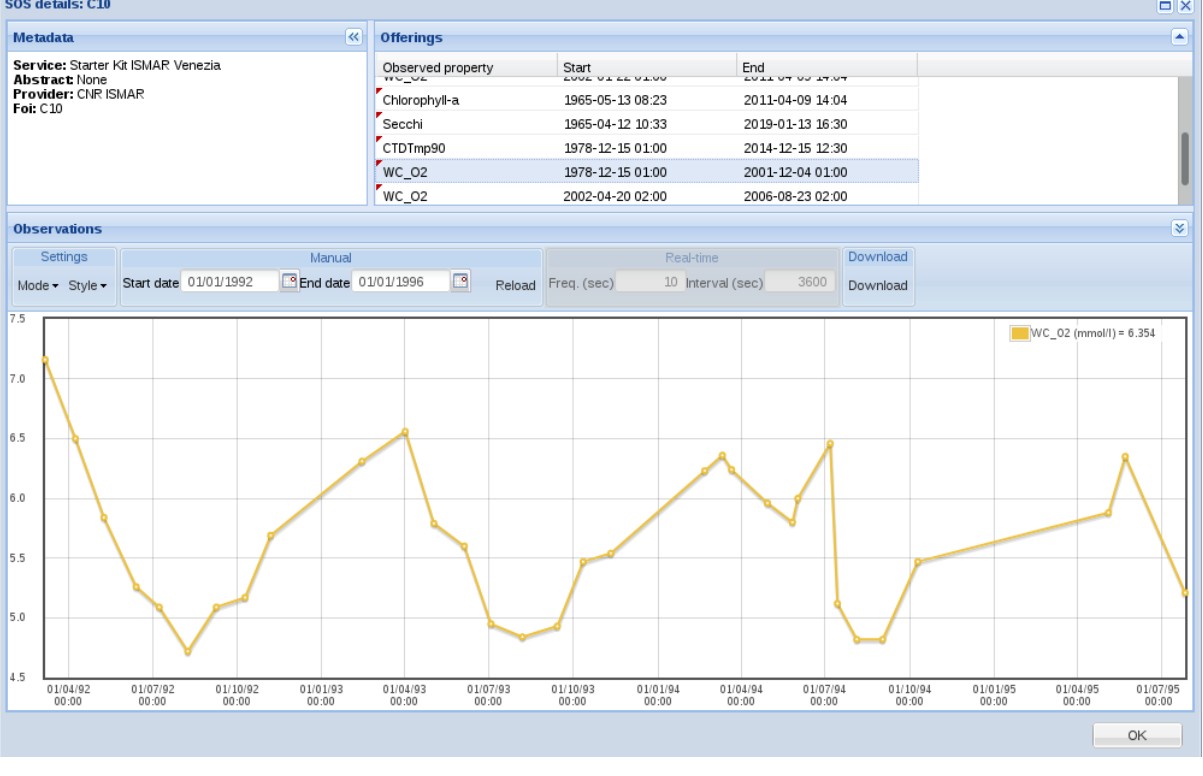

Figure 8 - Example of the graph in output from a query into the database. Oxygen data at station C10 for the period 01/01/1992-01/01/1996 are displayed.

Since v1.3.17 GET-IT still does not allow the three-dimensional representation of data, we decided to upload into the software suite only surface values of each parameter and sampling operation. This part of the database can be queried and graphed, directly into GET-IT using developed tool, in order to showtime series of selected parameters (Figure 8). A total of 16017 observations have been uploaded.

Observations can upload using the graphical interface or, for the skilled people, using an XML language directly into SOS (Sensor Observation Service) web service. For the upload from the interface, data have to be formatted in a table with datetime and parameter value (Figure 9). Since the speed of the process largely depends on the browser used to upload data, most of the data have been uploaded, through a Python script, by formatting specific .xml files, containing information about the sensor's ID, sampling station, and date time and following SWE standard. In both cases, the data upload begins with the selection of the sensor we want to upload data from and, then, with the selection of the sampling station from the map, if already available, or by creating a new one.

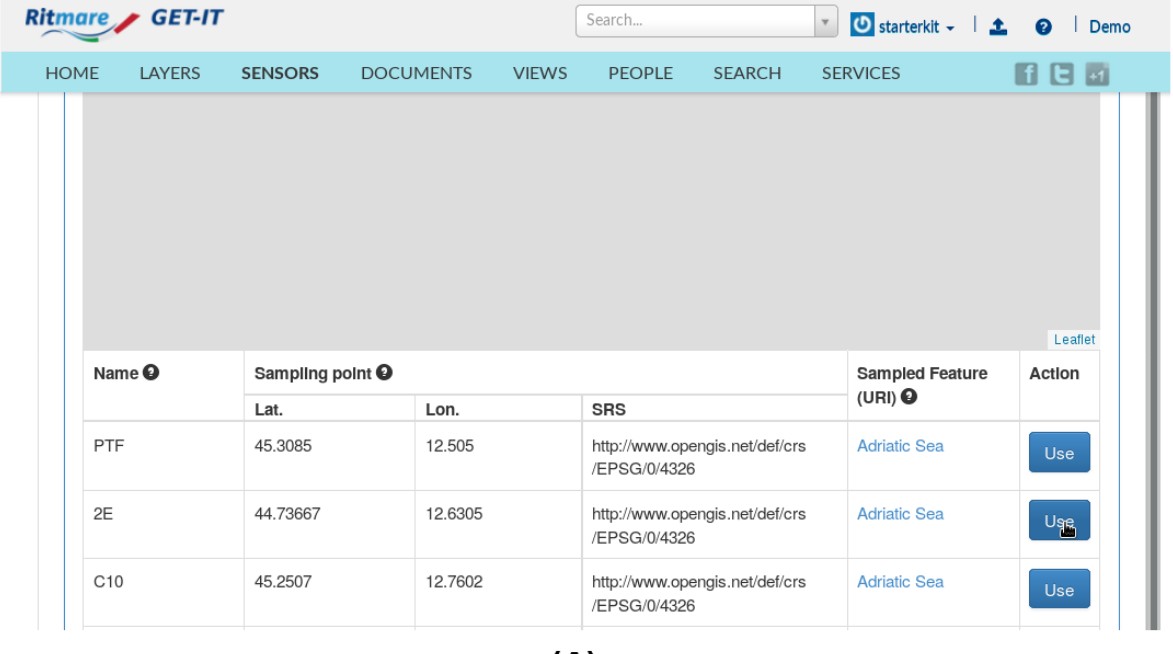

**(A)**

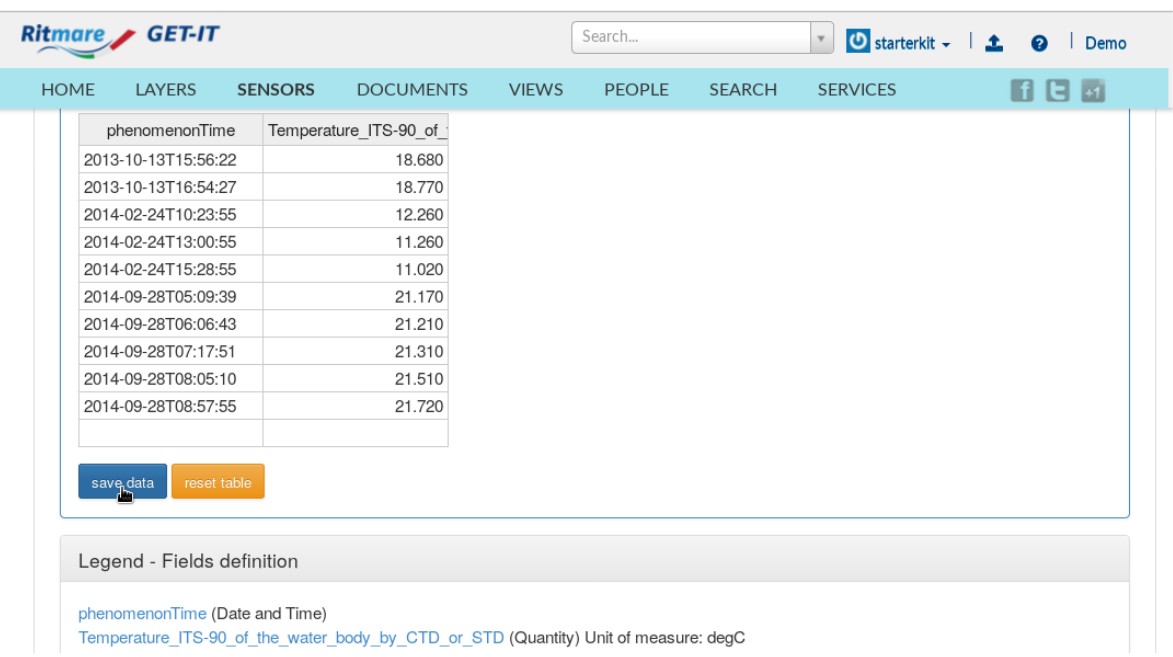

**(B)**

Figure 9 - Data upload from the graphic interface. Selection of the sampling station for the specific sensor (A) and format of
data to be uploaded into the SDI (B).

**6. Data availability**
The dataset is available at http://doi.org/10.5281/zenodo.3516717 (Acri et al., 2019). It was also uploaded in the DEIMS-SDR
repository (Dynamic Ecological Information Management System - Site and Dataset Registry,
https://deims.org/dataset/38d604ef-decb-4d67-8ac3-cc843d10d3ef), which is the official sites and data registry for LTER
International network. The aim of DEIMS-SDR is to be a catalogue of in-situ observation or experimentation facilities; it is
implemented as a web-based information portal for integrated ecological information which comprises detailed descriptions
of sites where research is carried out, including the technical infrastructure, ecosystem properties and research activities (see
Wohner et al., 2019 for a full description). DEIMS-SDR provides a service which allows to associate a PID (Persistent
IDentifier) to the uploaded dataset. Thanks to an agreement between the eLTER Research Infrastructure and the EUDAT
Collaborative Data Infrastructure (CDI), the dataset is automatically available also in the B2Share catalogue

(https://b2share.fz-juelich.de/records/e8d57102fd194bde957407ca290ad06a) and, through this, in the EOSC (European Open
Science Cloud) and GEOSS (Global Earth Observation System of Systems) catalogues. Since we opted for CC-BY license our
data are immediately fully available for download and reuse upon citation, without embargo rules or any further limitations.
Table 3 collects the list of columns, short name and extended name of each parameter and ancillary field composing the
database.

| Column number | Parameter short name (database) | Parameter extended name |
|---|---|---|
| 1 | Long | Longitude [decimal degrees] |
| 1 | Long | Longitude [decimal degrees] |
| 2 | Lat | Latitude [decimal degrees] |
| 3 | Depth | Depth [m] |
| 4 | Station | Name of sampled station |
| 5 | Station_updated_name | Updated name of sampling station, if present |
| 6 | Cruise | Cruise |
| 7 | Ship | Ship |
| 8 | YYYY-MM-DD | Date |
| 9 | hh:mm:ss | Time |
| 10 | Bot. Depth [m] | Water column depth [m] |
| 11 | Secchi [m] | Transparency [m] |
| 12 | Temp_sensor | Temperature sensor |
| 13 | Temp | Temperature [°C] |
| 14 | Sal_sensor | Salinity sensor |
| 15 | Sal | Salinity [dimensionless] |
| 16 | Dens | Density Anomaly [kg m$^{-3}$] |
| 17 | pH_sensor | pH sensor |
| 18 | pH | pH [pH Units] |
| 19 | Oxyg_sensor | Oxygen sensor |
| 20 | Oxyg (ml/l) | Dissolved Oxygen concentration [ml l$^{-1}$] |
| 21 | Ox% | Dissolved Oxygen saturation [%] |
| 22 | NH3_sensor | Ammonia sensor |
| 23 | NH3 (microMol) | Ammonia [µm dm$^{-3}$] |
| 24 | NO2_sensor | Nitrite sensor |
| 25 | NO2 (microMol) | Nitrite [µm dm$^{-3}$] |

| 26 | NO3_sensor | Nitrate sensor |
|---|---|---|
| 27 | NO3 (microMol) | Nitrate [$\mu$m dm$^{-3}$] |
| 28 | Din (microMol) | Total Dissolved inorganic Nitrogen [$\mu$m dm$^{-3}$] |
| 29 | PO4_sensor | Phosphate sensor |
| 30 | PO4 (microMol) | Phosphate [$\mu$m dm$^{-3}$] |
| 31 | Si_sensor | Silicate sensor |
| 32 | Si (microMol) | Silicate [$\mu$m dm$^{-3}$] |
| 33 | Chla_sensor | Chlorophyll-*a* sensor |
| 34 | Chla (ug/l) | Chlorophyll-*a* concentration [$\mu$g l$^{-1}$] |
| 35 | Pheo (ug/l) | Phaeopigments concentration [$\mu$g l$^{-1}$] |
| 36 | Alky | Alkalinity |
| 37 | Diato (cell/ml) | Diatoms abundance [cell ml$^{-1}$] |
| 38 | Dino (cell/ml) | Dinoflagellates abundance [cell ml$^{-1}$] |
| 39 | Flag (cell/ml) | Other Flagellates abundance [cell ml$^{-1}$] |
| 40 | Cocco (cell/ml) | coccolithophorids abundance [cell ml$^{-1}$] |
| 41 | Phyto TOT (cell/ml) | Total phytoplankton abundance [cell ml$^{-1}$] |
| 42 | Zoo (ind/m^3) | Total mesozooplankton organisms [ind m$^{-3}$] |

Table 3 - Correspondence between column number, short name and extended name of each parameter reported into the database.

**7. Conclusions**

The 50-year database of plankton and abiotic parameters in the NAS may contribute to an in-depth comprehension of plankton dynamics required not only to manage aquatic resources but also to predict and tackle future environmental changes. Long-term site-based studies on plankton may provide an invaluable opportunity to assess common or contrasting patterns of variability, to understand how those patterns change at different scales and to hypothesize about causes and consequences. Wide availability of the data on long-term variations of the planktonic system allows large scale studies that obviously go beyond the local use, representing a source of information for cross-system analysis, allowing comparison among ecosystems as well as new approaches in data analysis and in the development of water quality indicators.

However, these potential uses appear constrained by issues that are intrinsic to long-term series and that are related to the obvious variations, across the years, of sampling coverage and frequency and of analytical methodologies. In this respect, it is crucial to appropriately document the data, collecting and making available most ancillary information as possible on the changes occurred in time for each parameter measurement. This process was thoroughly carried for the 50 years NAS dataset so that the potential users might know which could be the proper application and the limitations of the dataset.

The open access to the 50-year dataset of abiotic data and plankton in the NAS was framed in a wider open science life-cycle approach undertaken in the EcoNAOS project (Minelli et al., 2018), with the purpose to develop a practical case study which could root the high and inspiring principles of Open Science into the scientific community, fostering as well a cultural shift. In EcoNAOS we involved, since its start, both LTER and data management researchers in a joint partnership. In particular, the elaboration of the 50-year datasets has been worked out by a small group of plankton ecologists and data management experts, with the aim of sharing and harmonizing as well the different experiences, needs and points of view. This participatory process is recognized to be crucial to contribute overcoming cultural differences, barriers and fragmentation that might represent an obstacle for Open Science (Björk, 2004; Janssen et al., 2012; Barry & Bannister, 2014). The constant interactions of

oceanographers and ecologists with experts on data management and analysis, geospatial standards and web services interoperability, creating a rich and multi-domain research group, has been necessary to make available and understandable the very detailed knowledge behind environmental surveys, samplings, analyses, methodologies, through the sound and fit-for-purpose technical solutions for data management and interoperability.

Accessibility and interoperability concepts and practices are crucial elements for LTER networks because the more the time series are consistent, coherent and available, the more it is possible to reconstruct trends and dynamics and to identify and compare reliable trends. The consistency and the coherence of the dataset require careful efforts in supplying the proper metadata, which could document the methodological changes that occurred through the years, thus allowing the potential users to evaluate the restrictions as well as the most suitable uses of the dataset. The activity described in this data paper is fully in line with the data management plan of the LTER networks, at the national, European and global level, since one of the LTER mandates is actually to foster open sharing of LTER data (Mirtl 2010; Mirtl et al., 2018). The national LTER networks are fostered to adopt the aspects of open science that are currently feasible in the different research groups.

Currently, a dynamic update and integration of the published dataset is not yet supported by specific tools nor integrated in automatic procedures; anyway, it is foreseen to go on with the promotion of a full open science approach to LTER also in the coming years and extend the dataset through the publication of updates and possibly through the integration of different long-term datasets.

## Author contribution

Minelli A. led the whole process of dataset recovery, metadatation and publication. Oggioni A., Pugnetti A. and Sarretta A. were the reference persons for the entire activity. Acri F., Bastianini M., Bernardi Aubry F., Camatti E., Bergami C., Minelli A., Oggioni A., Sarretta A., Pugnetti A. contributed to the writing of this paper: drafting of the text and of the figures and tables, revisions and suggestions.

Acri F., Bastianini M., Bernardi Aubry F., Camatti E., Boldrin A., Cassin D., De Lazzari A., Pansera M., Finotto S., Socal G. collected and analysed data over time, providing statistics and material (graphs and tables) for the paper.

## Competing interests

Authors declare that there are no competing interests that might have influenced the performance or presentation of the work described in this manuscript.

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
