# Peer review of "A long term (1965-2015) ecological marine database from the LTER-Italy site Northern Adriatic Sea: plankton and oceanographic observations"

_Earth System Science Data, 2019_

## Referee Comment (RC1) · Diego Fontaneto (Referee) · 2 Aug 2019

The datapaper is surely of great interest. Its positive sides are the presence of biotic and abiotic variables, the long-term coverage, and the detailed description of the rationale and the history of the data collection.

I think it will be highly relevant, even if it deals with a rather restricted area: the Northern Adriatic is one of the most studied seas of the world, due to the problems of Venice and high waters. Thus, the geographically restricted focus of the dataset is a positive

aspect, given the high scientific and political interest on the area.

I have only one major suggestion: at the moment, the manuscript does not describe in details the columns of the dataset. For example, the reader has to (easily) guess that Temp means temperature and Sal means salinity, but for other variables, such as Bot., Din, Si, Pheo, Alky, etc., it is not so easy and unambiguous to understand their meaning. It would be good to have the manuscript listing the headings of all the 43 columns of the dataset, and clearly explain what they are. For example, Table 2 could have an additional column unambiguously reporting the name of the column in the dataset for each of the variables. Such addition would make the dataset more relevant for more users.

Other minor issues that need to be addressed are:

Acronyms are not always explained: NAS in the abstract is not unambiguously reported, CTD on line 121, O&M on line 264, RDF on line 268, etc.

Figure 1 has lines in the water that are not explained in the figure caption (also in figure 3). Moreover, there is ambiguity between TeleSenigallia (in the figure) and Senigallia (in the caption).

Table 2 should explain the difference between "dimensionless" for salinity and "-" for pH.

Figure 4 does not contain exactly what is mentioned on line 215-221: only one column is present for "station", whereas line 217 states "Original station name and updated name", for which two different columns are expected from the reader; actually, the dataset has those two columns, but the figure does not have them. Both "sampling depth" on line 216 and "water column depth" on line 220 are mentioned in the text, but only one column with "depth" is present in figure 4.

There are some minor issues with the English language. Fore example the ambiguity between the terms "factor" and "parameter"; "applying of"; "much lowest"; "which" and

"that" not used consistently for restrictive and non-restrictive clauses (e.g. "which" on line 133 and 137, but "that" on line 305 and 307); "allow to" followed by a verb without a noun; etc.

---

## Referee Comment (RC2) · Diego Fontaneto (Referee) · 8 Aug 2019

Thanks for including all my suggestions. All the best with the manuscript.

---

## Author Comment (AC1) · 8 Aug 2019

Dear Dr. Fontaneto,

many thanks for the valuable and reasonable observations.

For the major suggestion you evidenced, we substantially agree with you and we will provide the extended name of each parameter observed by adding a column to the table 2, as you proposed. Moreover, we will add another, more comprehensive table in paragraph 6, reporting the extended name of all the observed and accessory

parameters (such as date-time, sensors name, coordinates).

For the minor issues you reported:

* we will correct the manuscript explaining all the acronyms in the order they appear in the revised version of the manuscript that we will upload after the discussion phase;

* we will correct English wording;

* lines in the water in figure 1 and 3 are territorial water limits (continuous violet lines) and navigation routes (dotted violet lines) contained in the original Open Street Map layer we used as a base map. Since they are not relevant to the figures or the context, we will provide a cleaned image (without these lines) in the revised version of the manuscript;

* for the Senigallia - Telesenigallia ambiguity in Figure 1: the transect is called "Senigallia-Susak", so there is an error in the figure that we will correct (Telesenigallia Pylon instead is a pylon belonging to the CNR fixed sensors observation network that records data in continuous, near Senigallia;

* in the database salinity is dimensionless, for pH we used the pH units. So, the "-" for pH record in Table 2 will be changed to "pH Units";

* effectively there is a difference between the caption of the database (Figure 4) and the description in lines 215-221. We will proceed to upload a corrected version of Figure 4 complying with the description provided in the revised version of the manuscript;

* we will correct factor/parameter ambiguity using only "parameter" term.

---

## Referee Comment (RC3) · Johan Wikner (Referee) · 11 Sep 2019

General comments The paper present a database composed of older data on paper and newer data recorded on spreadsheets and digital median for the northern Adriatic Sea. The database contains both hydrographic, chemical and biological data. Data collected comes from various research projects and monitoring programs. Consequently, sampling coverage, frequency and methodology used vary in the data set. The authors stress an aim to meet the requirements of Open Science and the FAIR data policy. Compilation of older data to extend long ecological times series back in time is very

valuable for both anthropogenic disturbance and climate change influence. In addition, to foster open and ready access to these data sets will promote general use and resue of already collected data. Some information about access to method protocols used would be informative. Also uncertainty statistic and quality codes indicating quality assurance level of data. Application of taxonomic lists for phyto- and zooplankton is an important aspect of this. Or comments on the lack of those. A table presenting characteristics of the main study areas would give a good overview. A more critical discussion regarding the potential of heterogenous time series to answer current ecological issue is requested. Any attempts to do so and the experience gained can be mentioned if available. Some information about the possibility to enter new data to the database and measures taken to promote homogenous and quality assured data should be added. Detailed comments r. 19 Please define "NAS" at first instance. r.46-47 Has all these advantages been convincingly documented? In what context have they been validated if so? r. 97 To me the Adriatic Sea is part of the Mediterranean. Please specify better. r. 102-103 It appears peculiar that a period of eutrophication and oligotrophication are referenced by citations only 1 year apart? Is that correct and may the periods in question be specified in that case. r. 108-109 A table with the variables in question would be valuable to get a quick overview. r. 110-111 A table describing the key characteristics of the investigated Sea areas would be valuable. r. 119 Is there and method protocol ID associated with the records for the different variables? Table 2 .Please provide name of the pH and oxygen sensor. CTD only covers Conductivity, Temperature and Density. An indication of depth coverage and sampling frequency range would be informative. Pleases use $\mu$mol dm-3 (or kg-1, for clarity and consistency). r. 142. Table 1 rather seem to show the record for specific research vessels rather than expeditions (cruises). Please rephrase the table legend. r. 182. Is there any quality codes presented for the records to assess the level of quality assurance performed? Are any uncertainty statistics or method declarations available? r. 204 I don't understand "handing down" please rephrase. Do you mean internal education and recurring inter-calibration of taxonomic competence? r. 206 What taxonomic lists are used to ensure consistent classification

of phyto- and zooplankton? r. 215 Both station coordinates and actually sampled coordinates should be used. Preferably with a 0.2 nm tolerance of station area. r. 227-228 It should be illustrative to give the distribution of samples divided in also chemistry and biology at least. r. 301-302. This is already presented at r. 116 and r. 26-27.Please harmonize and reduce text. r. 316-317. However, as pointed out variation in sampling coverage, frequency and methodology may impose constraints ion this potential. Please consider a more critical evaluation of the ability of the database to r. 335-337 Right, but comparability of methodology and data format also important to compare time series across sea areas (e.g. between countries) and even between aquatic environments . r. 340-341 Are there any limitation regarding access to the data that need to be mentioned? Or do the B2Share, EODC and GEOSS system provide full access to the data? Any quarantine rules of data that apply? Conclusion: Here or elsewhere you may present if new data can be entered in the database and what measures that are taken to improve data entry both manually and by import of files.

---

## Author Comment (AC2) · 24 Sep 2019

- r. 19: We will modify line 17 as follows: "In this paper, we describe a 50 years (1965-2015) ecological database containing data collected in the Northern Adriatic Sea (NAS)"

- r. 46-47: We will modify the paragraph as follows, adding some references: "From the researcher point of view, open practices have been reported to give advantage, first of all, to open new frontiers in science (Science|Business Network's Cloud Consultation

[Figure]

Group, 2019) and provide solutions to urgent societal problems (Palen et al., 2015; Tai and Robinson, 2018); moreover, it allows gaining more citations, media attention, potential collaborators, and funding opportunities (Eisenbach, 2006; McKiernan et al., 2016, Tennant et al., 2019) and it is vital for leaving a heritage to future generations"

References to be added:

* Science|Business Network's Cloud Consultation Group (2019). Why Open Science is the Future (and how to make it happen). Science|Business. Brussels. Report available here: https://sciencebusiness.net/report/why-open-science-future-and-how-make-it-happen

* Palen, L., Soden, R., Anderson, T. J., & Barrenechea, M. (2015, April). Success & scale in a data-producing organization: The socio-technical evolution of Open-StreetMap in response to humanitarian events. In Proceedings of the 33rd annual ACM conference on human factors in computing systems (pp. 4113-4122). ACM.

* Tai, T., & Robinson, J. (2018). Enhancing climate change research with open science. Frontiers in Environmental Science, 6, 115.

* Eysenbach, G. (2006). Citation advantage of open access articles. PLoS biology, 4(5), e157. DOI: 10.1371/journal.pbio.0040157

* Tennant JP, Crane H, Crick T, Davila J, Enkhbayar A, Havemann J, Kramer B, Martin R, Masuzzo P, Nobes A, Rice C, Rivera-López BS, Ross-Hellauer T, Sattler S, Thacker P, Vanholsbeeck M. 2019. Ten myths around open scholarly publishing. PeerJ Preprints 7:e27580v1 DOI: 10.7287/peerj.preprints.27580v1

- r. 97: Yes, Adriatic Sea is part of the Mediterranean area, we will modify the sentence as follows: "and the notable sea-level range, relatively to the rest of the Mediterranean area. . ."

- r. 102-103: We will modify the text, better specifying the periods of trophic changes and relative references: "The basin has undergone overfishing (Fortibuoni et al., 2010),

marked eutrophication (during the 70s; Giani et al., 2012), followed by a phase of oligotrophication (years 2000s; Mozetic et al., 2010) and by a recent increase of nutrient concentrations (since 2007; Totti et al., 2019)."

Reference to be added:

Giani et al., 2012. Recent changes in the marine ecosystems of the northern Adriatic Sea, Estuarine, Coastal, and Shelf Science, Volume 115, 2012, Pages 1-13, ISSN 0272-7714, https://doi.org/10.1016/j.ecss.2012.08.023.

- r. 108-109: The authors agree in referencing Table 2 here since it gives a complete overview of all the parameters examined. We will modify the sentence as follows: "The LTER-Italy parent site NAS includes four research sites (Gulf of Trieste, Gulf of Venice, Po Delta and Romagna Coast, Senigallia-Susak Transect; Figure 1), where meteo-oceanographic and biological data, mainly on plankton (Table 2), are gathered both during oceanographic cruises and at fixed point observatories."

- r. 110 -111: The dataset we describe here refers to the whole NAS, which includes also the 4 LTER research sites but is a much wider area, described in detail in the text (lines 93-112). The Authors believe that additional descriptions only of the four research sites could be a little bit misleading.

- r. 119: We will better explain the level of metadatation and accessibility of data by adding the following sentence after line 130: "In particular, methodological protocols and associated documentation changed through time. t Several sensors are described and extensively documented through the GET-IT platform (see Paragraph 5), where it is possible to visualize all the observations related to a specific instrument or method. Other protocols have undergone a deep metadatation process by analyzing ancillary historical metadata (Scovacricchi, 2017). In this case, it is not immediately possible to obtain data related to a specific protocol, but it is still possible to filter data by method by importing the .csv file in a spreadsheet."

ps: the partial upload of data through GET-IT platform is justified in rows 270-288.

- Tab. 2:

\* We will add to the table the name of pH sensors (pH glass membrane and pH electrode) and Oxygen sensor (Oxygen Polarographic sensor).

\* For the indication of depth coverage and sampling frequency range we will add this sentence in paragraph 3 (rows 121-123): "Sampling frequency: e.g., data coming from CTD, such as temperature, oxygen, and pH, are registered in real-time at each meter in depth; other parameters, like nutrients and phytoplankton, are sampled at a much lowest time-frequency and at variable depths. The depth coverage ranged between 0-63 m, the sampling frequency from monthly to seasonal"

\* We will change the measurement unit ($\mu$m dm-3 ) of nutrients as requested

- Tab. 1: We will change the caption as follow: "Operation periods of the different research vessels between 1965 and 2015 and number of observations"

- r. 182: We will add the following sentence at rows 182-183 in order to clarify the level of quality assurance of data: "Samples collected during each cruise, whatever the station, were then analyzed in the laboratory by means of diverse techniques. Since 2000 analytical quality of nutrients and chlorophyll analyses is assessed through participation to the Quality Assurance of Information for Marine Environmental Monitoring In Europe (QUASIMEME; http://www.quasimeme.org) international laboratory proficiency-testing. The complete . . ."

- r. 204: We will modify the sentence as follows: "To deal with this issue, internal education and recurring calibration of taxonomic competence were carefully considered, with training periods and intercalibrations phases."

- r. 206: Here we will add the following sentence in order to complete the list of taxonomic references we adopted: "Since 2006 the taxonomic revision of the phytoplankton species has been made according to the global algal database of taxonomic, nomenclatural and distributional information "Algaebase" (www.algaebase.org), for the zooplankton the Marine Planktonic Copepods catalog (https://copepodes.obs-banyuls.fr/en/links.php, Razolus et al., 2005-2019) has been used. In the past, for phyto and zooplankton analyses several texts and monographs were used (Berard-Therriault et al., 1999; Harris et al., 2000; Heimdal, 1993; Hendey, 1964; Hustedt, 1930-1966; Pascher, 1915; Peragallo and Peragallo, 1897-1908; Rampi and Bernhardt, 1980; Schiller, 1931-37; Throndsen, 1993; Tomas, 1997)"

References to be added:

* Razouls C., de Bovée F., Kouwenberg J. and Desreumaux N., 2005-2019. - Diversity and Geographic Distribution of Marine Planktonic Copepods. Sorbonne University, CNRS. Available at http://copepodes.obs-banyuls.fr/en [Accessed September 24, 2019]

* Berard-Therriault L., Poulin M., Bossé L. 1999. Guide d'identification du phytoplancton marin de l'estuaire et du golfe du Saint-Laurent. NRC Research Press, 387 pp.

* Harris, R.P., Wiebe, P.H., Lenz, J., Skjoldal, H.R. and M. Huntley. 2000. ICES Zooplankton Methodology Manual, Academic Press, USA. pp. 684

* Heimdal B. R., 1993 Modern Coccolithophorids in: Marine phytoplankton a guide to naked flagellates and coccolithophorids. Tanos editors, Academic Press: 147- 248.

* Hendey, N. I., 1964. An introductory account of the smaller algae of British coastal waters. Part V: Bacillariophyceae, Diatoms. Fishery Invest. Lond. Ser. IV 5, 317 pp.

* Hustedt F., 1930-1966. Die Kiesealgen von Deutschland, Österreichs und der Schweiz mit Berusichtigung der übrigen Länder Europas sowie der angrenzender Mehresgebiete. In : Rabenhorst's Kriptogamen-Flora von Deutschland, Österreichs und der Schweiz. Akad; Verlag. m. b. H. Leipzig. 7 : Tl. 2. 920 pp. : Tl., 2 845 pp. ; Tl. 3, 816 pp.

* Pascher A. 1915. Clorophyceae. In: Die Susswasser Flora Deutschlands, Osterreichs und der Schweiz. Verlags von Gustav Fisher, Jena, Heft 5, 250 pp.

* Peragallo H, Peragallo M., 1897-1908. Diatomees Marine de France et des Districts Maritimes Voisins. Micrographe Editeur Grez sur Loing (S. et M.), 419 pp.

* Rampi L., Bernhardt M., 1980. Chiave per la determinazione tassonomica delle Peridinee Pelagiche Mediterranee: C.N.E.N., Roma (RT/B10 (81)13): 1-98.

* Schiller J., 1931-37. Dinoflagellatae (Peridineae) Monografischer Behandlung. In : Rabenhorst Kriptogamen-Flora von Deutschland, Österreichs und der Schweiz. Verlag. m. b. H. Leipzig. 10 (3) -1, 1-617, (1931-1933), (10) 3-2, 1-590, (1933-1937).

* Sournia A., 1993. Atlas du phytoplancton marin. Editions du Centre National de la recherche Scientifique. (1), 1-219, (2) 1-297.

* Throndsen J., 1993. The planktonic marine flagellates in: Marine phytoplankton a guide to naked flagellates and coccolithophorids. Tanos editors, Academic Press: 7-131.

* Tomas, C. R., 1997. Identifying Marine Phytoplankton. Academic Press, Arcourt Brace & Company.

- r. 215: We decided not to add coordinates of standard sampling stations to the database because the substantial validity of these stations is limited to the period prior to the advent of GPS on board. In fact, after 90s standard sampling stations started to lose their significance since station names were no more related to the name of the station but to coordinates themselves. Furthermore, the coordinates of standard sampling stations used as a reference in sampling until 90s are available as "stationsAll.csv" file via GitHub at the following link: https://github.com/CNR-ISMAR/econaos/tree/master/sampleData

- r. 227-228: The authors decided to individuate as abiotic parameters nutrients, alkalinity, and transparency. Biotic parameters are chlorophyll, pheo-pigments, phytoplankton, and zooplankton. In the updated version of the data paper, we will add a graph to

Figure 5 indicating their trend over the 50 years.

- r. 301-302: We will delete the link to the dataset at row 116, but the link to the database is mandatory for the journal both in the abstract and in the "data availability" section. This prescription is reported in the ESSD guidelines for authors.

- r. 316-317 e r335-337: We will add some sentences to evidence the importance to collect metadata covering more than the possible all the observations, in particular:

* r. 322: "..in the development of water quality indicators. However, these potential uses appear constrained by issues that are intrinsic to long-term series and that are related to the obvious variations, across the years, in sampling coverage and frequency and in analytical methodologies. In this respect, it is crucial to appropriately document the data, collecting and making available most ancillary information as possible on the changes occurred in time for each parameter measurement. This process was thoroughly carried for the 50 years NAS dataset so that the potential users might know which could be the proper application and the limitations of the dataset."

* r. 336 "..to identify and compare reliable trends. The consistency and the coherence of the dataset require efforts in supplying the proper metadata, which could document the methodological changes that occurred through the years, thus allowing the potential users to evaluate the restrictions as well as the most suitable uses of the dataset."

- r. 340-341: For the limitation to data access we will modify and add a sentence in the "data availability" section (r. 310) : "Thanks to an agreement between the eLTER Research Infrastructure and the EUDAT Collaborative Data Infrastructure (CDI), the dataset is automatically available also in the B2Share catalog (http://hdl.handle.net/21.11125/4672def7-4aeb-47e0-a325-311d02860967) and, through this, in the EOSC (European Open Science Cloud) and GEOSS (Global Earth Observation System of Systems) catalogs. Since we opted for CC-BY license on our data, data is immediately fully available for download and reuse upon citation. There are no embargo rules or any further limitations in this specific case.

For new data entry, we described in the conclusions paragraph a possible envisaged approach as follows (r-341): "Currently, a dynamic update and integration of the published dataset is not yet supported by specific tools nor integrated in automatic procedures; anyway, it is foreseen to go on with the promotion of a full open science approach to Long Term Ecological Research also in the coming years and extend the dataset through the publication of updates and possibly through the integration of different long-term datasets."

---

## Author Comment (AC3) · 25 Sep 2019

Please consider this response for what concerns correction demanded at row 227-228 instead of the one given in the previous Author Comment

- r. 227-228: In the updated version of the data paper, we will add a graph to Figure 5 indicating the trend over the 50 years of abiotic (nutrients, alkalinity, and transparency) and biotic parameters (chlorophyll, phytoplankton, and zooplankton).

[Figure]

2019.

---

## Author Response (AR1)

*Modifications envisaged and supplemental comments in 2nd version of the manuscript, following 1st referee (Diego Fontaneto) suggestions:*

For the major suggestion you evidenced, we substantially agree with you and we provided the extended name of each parameter observed by adding a column to the table 2 as you propose. Moreover, we added another, more comprehensive table in paragraph 6, reporting the extended name of all the observed and accessory parameters (such as date time, sensors name, coordinates).

For the minor issues you reported:
- we corrected the manuscript explaining all the acronyms in the order they appear in the revised version of the manuscript that we will upload after the discussion phase;

- we corrected english wording, as you suggested;

- lines in the water in figure 1 and 3 are territorial water limits (continuous violet lines) and navigation routes (dotted violet lines) contained in the original Open Street Map layer we used as a basemap. Since they are not relevant to the figures or the context, we provided a cleaned image (without these lines) in the revised version of the manuscript;

- for the Senigallia - Telesenigallia ambiguity in Figure 1: the transect is called "Senigallia-Susak", so there is an error in the figure that we corrected (Telesenigallia Pylon instead is a pylon belonging to the CNR fixed sensors observation network that records data in continuous, near Senigallia);

- in the database salinity is dimensionless, for pH we used the pH units. So, the "-" for pH record in the Table 2 has been changed to "pH Units";

- effectively there is a difference between the caption of the database (Figure 4) and the description in lines 215-221. We uploaded a corrected version of the Figure 4 complying with the description provided in the revised version of the manuscript;

- we corrected factor/parameter ambiguity using only "parameter" term.

*Modifications envisaged and supplemental comments in 2nd version of the manuscript, following 2nd referee (Johan Wikner) suggestions:*

- r. 19: We modified line 17 as follows:
"In this paper, we describe a 50 years (1965-2015) ecological database containing data collected in the Northern Adriatic Sea (NAS)"

- r. 46-47: We modified the paragraph as follows, adding some references:
"From the researcher point of view, open practices have been reported to give advantage, first of all, to open new frontiers in science (Science|Business network's cloud consultation group, 2019) and provide solutions to urgent societal problems (Palen et al., 2015; Tai and Robinson, 2018); moreover, it allows gaining more citations, media attention, potential collaborators, and funding opportunities (Eisenbach, 2006; McKiernan et al., 2016, Tennant et al., 2019) and it is vital for leaving a heritage to future generations."

References added:

- Science|Business Network's Cloud Consultation Group (2019). Why Open Science is the Future (and how to make it happen). Science|Business. Brussels. Report available here: https://sciencebusiness.net/report/why-open-science-future-and-how-make-it-happen
- Palen, L., Soden, R., Anderson, T. J., & Barrenechea, M. (2015, April). Success & scale in a data-producing organization: The socio-technical evolution of OpenStreetMap in response to humanitarian events. In Proceedings of the 33rd annual ACM conference on human factors in computing systems (pp. 4113-4122). ACM.
- Tai, T., & Robinson, J. (2018). Enhancing climate change research with open science. Frontiers in Environmental Science, 6, 115.
- Eysenbach, G. (2006). Citation advantage of open access articles. PLoS biology, 4(5), e157. DOI: 10.1371/journal.pbio.0040157
- Tennant JP, Crane H, Crick T, Davila J, Enkhbayar A, Havemann J, Kramer B, Martin R, Masuzzo P, Nobes A, Rice C, Rivera-López BS, Ross-Hellauer T, Sattler S, Thacker P, Vanholsbeeck M. 2019. Ten myths around open scholarly publishing. PeerJ Preprints 7:e27580v1 DOI: 10.7287/peerj.preprints.27580v1

- r. 97: Yes, of course the Adriatic Sea is part of the Mediterranean area, we modified the sentence as follows:
"and the notable sea-level range, relatively to the rest of the Mediterranean area…"

- r. 102-103: We modified the text, better specifying the periods of trophic changes and the references:
"The basin has undergone overfishing (Fortibuoni et al., 2010), marked eutrophication (during the 70s; Giani et al., 2012), followed by a phase of oligotrophication (years 2000s; Mozetič et al., 2010) and by a recent increase of nutrient concentrations (since 2007; Totti et al., 2019). The NAS has also been subjected to frequent development of mucilage aggregates (Giani et al., 2005; De Lazzari et al., 2008), until the first decade of the 2000s."

Reference added:

Giani et al., 2012. Recent changes in the marine ecosystems of the northern Adriatic Sea, Estuarine, Coastal, and Shelf Science, Volume 115, 2012, Pages 1-13, ISSN 0272-7714, https://doi.org/10.1016/j.ecss.2012.08.023.

- r. 108-109: The authors agree in referencing Table 2 here since it gives a complete overview of all the parameters examined. We modified the sentence as follows:
"The LTER-Italy parent site NAS includes four research sites (Gulf of Trieste, Gulf of Venice, Po Delta and Romagna Coast, Senigallia-Susak Transect; Figure 1), where meteo-oceanographic and biological data, mainly on plankton (Table 2), are gathered both during oceanographic cruises and at fixed point observatories."

- r. 110 -111: The dataset we describe here refers to the whole NAS, which includes also the 4 LTER research sites but is a much wider area, described in detail in the text (lines 93-112). The Authors believe that additional descriptions only of the four research sites could be a little bit misleading.

- r. 119: We better explained the level of metadatation and accessibility of data by adding the following sentence after line 130:
"In particular, methodological protocols and associated documentation changed through time. Several sensors are described and extensively documented through the GET-IT platform (see Section 5), where it is possible to visualize all the observations related to a specific instrument or method. Other protocols have undergone a deep metadatation process by analyzing ancillary historical metadata (Scovacricchi, 2017). In this case, it is not immediately possible to obtain data related to a specific protocol, but it is still possible to filter data by method by importing the .csv file in a spreadsheet."

ps: the partial upload of data through GET-IT platform is justified in rows 270-288.

- Tab. 2:
  ● We added to the table the name of pH sensors (pH glass membrane and pH electrode) and Oxygen sensor (Oxygen Polarographic sensor).
  ● For the indication of depth coverage and sampling frequency range we added this sentence in paragraph 3 (rows 121-123):
    "Sampling frequency: e.g., data coming from CTD (Conductivity, Temperature, Depth), such as temperature, oxygen, and pH, are registered in real-time at each meter in depth; other parameters, like nutrients and phytoplankton, are sampled at a much lowest time-frequency and at variable depths. The depth coverage ranged between 0-63 m, the sampling frequency from monthly to seasonal"
  ● We changed the measurement unit ($\mu m\ dm^{-3}$) of nutrients as requested

- Tab. 1: We changed the caption as follow:
"Operation periods of the different research vessels between 1965 and 2015 and number of observations"

- r. 182: We added the following sentence at rows 182-183 in order to clarify the level of quality assurance of data:
"Samples collected during each cruise, whatever the station, were then analyzed in the laboratory by means of diverse techniques. Since 2000 analytical quality of nutrients and chlorophyll analyses is assessed through participation to the Quality Assurance of Information for Marine Environmental

Monitoring In Europe (QUASIMEME; http://www.quasimeme.org) international laboratory proficiency-testing."

- r. 204: We modified the sentence as follows:
"To deal with this issue, internal education and recurring calibration of taxonomic competence were carefully considered, with training periods and intercalibrations phases."

- r. 206: Here we added the following sentence in order to complete the list of taxonomic references we adopted:
"Since 2006 the taxonomic revision of the phytoplankton species has been made according to the global algal database of taxonomic, nomenclatural and distributional information "Algaebase" (www.algaebase.org), the global algal database of taxonomic, nomenclatural and distributional information, for the zooplankton the Marine Planktonic Copepods catalog (https://copepodes.obs-banyuls.fr/en/links.php, Razolus et al., 2005-2019) has been used. In the past, for phyto- and zooplankton analyses several texts and monographs were used (Berard-Therriault et al., 1999; Harris et al., 2000; Heimdal, 1993; Hendey, 1964; Hustedt, 1930-1966; Pascher, 1915; Peragallo and Peragallo, 1897-1908; Rampi and Bernhardt, 1980; Schiller, 1931-37; Throndsen, 1993; Tomas, 1997)"

References added:

- Razouls C., de Bovée F., Kouwenberg J. and Desreumaux N., 2005-2019. - Diversity and Geographic Distribution of Marine Planktonic Copepods. Sorbonne University, CNRS. Available at http://copepodes.obs-banyuls.fr/en [Accessed September 24, 2019]
- Berard-Therriault L., Poulin M., Bossé L. 1999. Guide d'identification du phytoplancton marin de l'estuaire et du golfe du Saint-Laurent. NRC Research Press, 387 pp.
- Harris, R.P., Wiebe, P.H., Lenz, J., Skjoldal, H.R. and M. Huntley. 2000. ICES Zooplankton Methodology Manual, Academic Press, USA. pp. 684
- Heimdal B. R., 1993 Modern Coccolithophorids in: Marine phytoplankton a guide to naked flagellates and coccolithophorids. Tanos editors, Academic Press: 147- 248.
- Hendey, N. I., 1964. An introductory account of the smaller algae of British coastal waters. Part V: Bacillariophyceae, Diatoms. Fishery Invest. Lond. Ser. IV 5, 317 pp.
- Hustedt F., 1930-1966. Die Kiesealgen von Deutschland, Österreichs und der Schweiz mit Berusichtigung der übrigen Länder Europas sowie der angrenzender Mehresgebiete. In : Rabenhorst's Kriptogamen-Flora von Deutschland, Österreichs und der Schweiz. Akad; Verlag. m. b. H. Leipzig. 7 : Tl. 2. 920 pp. : Tl., 2 845 pp. ; Tl. 3, 816 pp.
- Pascher A. 1915. Clorophyceae. In: Die Susswasser Flora Deutschlands, Osterreichs und der Schweiz. Verlags von Gustav Fisher, Jena, Heft 5, 250 pp.
- Peragallo H, Peragallo M., 1897-1908. Diatomees Marine de France et des Districts Maritimes Voisins. Micrographe Editeur Grez sur Loing (S. et M.), 419 pp.
- Rampi L., Bernhardt M., 1980. Chiave per la determinazione tassonomica delle Peridinee Pelagiche Mediterranee: C.N.E.N., Roma (RT/B10 (81)13): 1-98.
- Schiller J., 1931-37. Dinoflagellatae (Peridineae) Monografischer Behandlung. In : Rabenhorst Kriptogamen-Flora von Deutschland, Österreichs und der Schweiz. Verlag. m. b. H. Leipzig. 10 (3) -1, 1-617, (1931-1933), (10) 3-2, 1-590, (1933-1937).
- Sournia A., 1993. Atlas du phytoplancton marin. Editions du Centre National de la recerche Scientifique. (1), 1-219, (2) 1-297.
- Throndsen J., 1993. The planktonic marine flagellates in: Marine phytoplankton a guide to naked flagellates and coccolithophorids. Tanos editors, Academic Press: 7- 131.

- Tomas, C. R., 1997. Identifying Marine Phytoplankton. Academic Press, Arcourt Brace & Company.

- r. 215: We decided not to add coordinates of standard sampling stations to the database because the substantial validity of these stations is limited to the period prior to the advent of GPS on board. In fact, after 90s standard sampling stations started to lose their significance since station names were no more related to the name of the station but to coordinates themselves. Furthermore, the coordinates of standard sampling stations used as a reference in sampling until 90s are available as "stationsAll.csv" file via GitHub at the following link: https://github.com/CNR-ISMAR/econaos/tree/master/sampleData

- r. 227-228: In the updated version of the data paper, we added a graph to Figure 5 indicating the trend over the 50 years of abiotic (nutrients, alkalinity, and transparency) and biotic parameters (chlorophyll, phytoplankton, and zooplankton).

- r. 301-302: We deleted the link to the dataset at row 116, but the link to the database is mandatory for the journal both in the abstract and in the "data availability" section. This prescription is reported in the ESSD guidelines for authors.

- r. 316-317 e r335-337: We added some sentences to evidence the importance to collect metadata to document the methodological changes occurred across the years, in particular:

- r. 322: "..in the development of water quality indicators. However, these potential uses appear constrained by issues that are intrinsic to long-term series and that are related to the obvious variations, across the years, of sampling coverage and frequency and of analytical methodologies. In this respect, it is crucial to appropriately document the data, collecting and making available most ancillary information as possible on the changes occurred in time for each parameter measurement. This process was thoroughly carried for the 50 years NAS dataset so that the potential users might know which could be the proper application and the limitations of the dataset."
- r. 336 "..to identify and compare reliable trends. The consistency and the coherence of the dataset require careful efforts in supplying the proper metadata, which could document the methodological changes that occurred through the years, thus allowing the potential users to evaluate the restrictions as well as the most suitable uses of the dataset."

- r. 340-341: For the limitation to data access we modified and add a sentence in the "data availability" section (r. 310) :
"Thanks to an agreement between the eLTER Research Infrastructure and the EUDAT Collaborative Data Infrastructure (CDI), the dataset is automatically available also in the B2Share catalogue (http://hdl.handle.net/21.11125/4672def7-4aeb-47e0-a325-311d02860967)(https://b2share.fz-juelic h.de/) and, through this, in the EOSC (European Open Science Cloud) and GEOSS (Global Earth Observation System of Systems) catalogues. Since we opted for CC-BY license our data are immediately fully available for download and reuse upon citation, without embargo rules or any further limitations."

For new data entry, we described in the conclusions paragraph a possible envisaged approach as follows (r-341):
"Currently, a dynamic update and integration of the published dataset is not yet supported by specific tools nor integrated in automatic procedures; anyway, it is foreseen to go on with the

promotion of a full open science approach to LTER also in the coming years and extend the dataset through the publication of updates and possibly through the integration of different long-term datasets."

*Comments in the marked-up manuscript lost in conversion to .pdf format:*

[1] We had to change the DOI reference since we published a new version of the database: in the first version, a column resulted to be duplicated, in the new version we deleted that column.
[2] Table added
[3] Table deleted
[4] We had to change the DOI reference since we published a new version of the database: in the first version, a column resulted to be duplicated, in the new version we deleted that column.

[revised manuscript text omitted]

---

## Author Response (AR2)

*Modifications made in the minor revision process:*

We had to change the DOI reference since we noticed, soon after the first revision process, that some values regarding the zooplankton parameter were incorrect. Specifically, we compared the published version of the database with another, older database version and we found discrepancies maybe due to an erroneous manual transcription from one version to the other. We corrected the database for those particular zooplankton values, we then re-checked the entire database in order to have a definitive version of our data and we published it on Zenodo. The database has now a new DOI (http://doi.org/10.5281/zenodo.3516717), which is reported in this revised manuscript both in the Abstract and in the Data availability section, as required.

[revised manuscript text omitted]